# An Irreversible and Revocable Template Generation Scheme Based on Chaotic System

**DOI:** 10.3390/e25020378

**Published:** 2023-02-18

**Authors:** Jinyuan Liu, Yong Wang, Kun Wang, Zhuo Liu

**Affiliations:** 1College of Computer Science and Technology, Chongqing University of Posts and Telecommunications, Chongqing 400065, China; 2School of Intelligent Technology and Engineering, Chongqing University of Science and Technology, Chongqing 401331, China; 3School of Mathematics and Big Data, Guizhou Education University, Guiyang 550018, China

**Keywords:** template protection, privacy protection, biometric security, chaotic system

## Abstract

Face recognition technology has developed rapidly in recent years, and a large number of applications based on face recognition have emerged. Because the template generated by the face recognition system stores the relevant information of facial biometrics, its security is attracting more and more attention. This paper proposes a secure template generation scheme based on a chaotic system. Firstly, the extracted face feature vector is permuted to eliminate the correlation within the vector. Then, the orthogonal matrix is used to transform the vector, and the state value of the vector is changed, while maintaining the original distance between the vectors. Finally, the cosine value of the included angle between the feature vector and different random vectors are calculated and converted into integers to generate the template. The chaotic system is used to drive the template generation process, which not only enhances the diversity of templates, but also has good revocability. In addition, the generated template is irreversible, and even if the template is leaked, it will not disclose the biometric information of users. Experimental results and theoretical analysis on the RaFD and Aberdeen datasets show that the proposed scheme has good verification performance and high security.

## 1. Introduction

Compared with fingerprint recognition and other biometric authentication methods, face recognition has been widely used in many fields because of its non-contact characteristics [1]. Due to the fact that the biometrics have characteristics that are difficult to change, once the face feature is leaked, it means permanent disclosure, which will pose a persistent threat to users [2]. Therefore, in recent years, more and more users have begun to pay attention to the security of face recognition systems. Because the template generated by the face recognition system is highly correlated with the feature information of user, it is very important to protect the template.

Many researchers have carried out research on template protection and believe that an ideal biometric template should have the following characteristics [3,4,5]:Irreversibility: The original face feature information cannot be derived from templates, or it is very difficult to calculate, so as to ensure that templates cannot be used for any purpose other than the original expectation.Revocability: Once the template is leaked, it can easily generate a new protected template. The leaked template has no utility in the newly generated template.Diversity/Unlinkability: Different templates can be generated for different applications based on the same biometric data of user. These different templates are not allowed to cross match between applications.Performance preservation: By using the template protection scheme, the original biometric information may have some loss, which will affect the accuracy of recognition. The template protection scheme should ensure accuracy.

However, templates face many security threats. Some template protection schemes have been successfully cracked, due to insufficient security, which brings huge security risks to registered users [6,7]. Therefore, this paper aims to design a secure template protection scheme to ensure the information security of registered users. Chaotic systems with excellent performance are highly sensitive to initial values [8], and the generated chaotic sequences have good randomness, so they have been widely used in random number generators, image encryption, and other fields in recent years [9,10]. Applying chaotic systems to feature template protection is beneficial to improve security and make the generation process more convenient. The proposed scheme uses the chaotic sequence generated by the chaotic system to drive the generation process, ensuring that the scheme has good revocability and diversity. After vector permutation and vector transformation, the extracted face feature vector is calculated with a random matrix, and the calculation results are converted into integers to generate a template. This process ensures the irreversibility and performance preservation of the templates. The scheme has good security, which can ensure that the template will not disclose the original feature information of registered users and can prevent the malicious use of the template. Here, we list the novelty and main contributions of this paper:The chaotic system is applied to the template protection, which enhances the revocability and diversity of the scheme.The orthogonal matrix is used to transform the feature vector, which minimizes the impact of the generation process on the performance of the template.Convert the cosine values of the feature vector and the random vectors into integers to generate template, making the scheme irreversible.Experiments on different datasets and theoretical analyses prove that the scheme is safe and efficient.

The rest of the article is organized as follows. Section 2 introduces some representative template protection schemes. Section 3 describes the proposed template protection scheme and its advantages. Section 4 analyzes the effectiveness and safety of the proposed scheme. Finally, Section 5 summarizes this paper.

## 2. Related Work

In a typical face recognition system, the user inputs a face image at the registration stage, and the system generates a template and stores it in the database. In the query stage, the system generates the query information according to the query image, then extracts the template from the database for matching and returns the comparison result. Storing templates in an insecure way will increase the risk of being attacked and data leakage [11]. Many researchers are committed to the study of template protection. Among them, feature transformation and feature encryption are the two most widely studied methods [12].

Feature transformation uses transform functions to protect the extracted face vector. According to the type of transformation function used, it can be divided into reversible transformation and non-invertible transformation [13]. Hash is one of the commonly used feature transformation methods. Jin et al. [14] proposed a template protection method based on Index-of-Max (IoM) hash. IoM hash converts biometric vectors into discrete index hash codes through externally generated random parameters. The template generated by this method has strong concealment and robustness to biometric changes. However, this scheme is vulnerable to authentication and cross-link attacks [6]. Dang et al. [15] proposed a full entropy hash algorithm for face template protection. The algorithm encodes the original biometric data into a hash value and uses the hash value as the template. The hash value generated by the algorithm has good randomness, distinguishability, and non-linkability. Alwan et al. [16] proposed a template generation algorithm based on the winner-takes-all hash. This algorithm transforms the extracted face feature vector with random binary orthogonal matrix and then uses the winner-takes-all hash to match. Because the mathematical formula of the algorithm is complex and the calculation amount is large, it is not suitable for real-time face verification.

Random perturbation is another commonly used feature transformation method. Kang et al. [17] proposed a two-factor face authentication scheme based on matrix transformation and user key. The template is generated by perturbing the feature vector through the matrix and the template can be changed freely. However, this scheme uses the general invertible matrix as the permutation matrix. If the user key is stolen, the security of the template will not be guaranteed. Nakamura et al. [18] proposed a template generation scheme based on unitary transformation. The Euclidean distance of different vectors is the same as that of the original vector after being perturbed by unitary transformation, so this scheme has good recognition performance. In addition, the template generated by this scheme can be republished multiple times without original information. However, this scheme is the same as the previous one, and its security depends on the confidentiality of parameters. Kumar et al. [19] proposed a local preserving projection method based on random perturbation and applied it to template protection. To ensure the difference of templates of different users, the scheme generates a unique personal identification code for each user. The random disturbance matrix is determined by the identification code. However, the personal identification code is required for user authentication. This limits the practicability of the scheme. Manisha et al. [20] proposed a reversible feature template generation scheme combining random perturbation and Chinese remainder theorem. In this scheme, the original image is randomly perturbed by the mask image to preserve the intensity of the feature, and then the Chinese remainder theorem is used to ensure the privacy of the intensity value. In extreme cases, for example, if the attacker obtains the template database and the mask image at the same time, and the attacker’s registration information is in the database, the scheme is insecure.

In feature encryption methods, the biometric feature is encrypted and used as template. Feature encryption methods mainly include key generation [21] and key binding [22]. Faragallah et al. [23] used Baker mapping to generate biometric templates. Different convolution kernels are generated by Baker mapping to generate templates, which effectively improves the diversity of the schemes. The scheme can perform verification in the encryption domain and avoid the risk of information leakage when the template is decrypted for verification. Dong et al. [24] proposed a face recognition scheme that only requires biometric input. The scheme consists of a one-to-many search subsystem and a one-to-one fuzzy matching subsystem. During user verification, the former returns k approximate matching objects through the maximum index hash, while the latter accurately matches k objects. The scheme has advantages in accuracy, calculation cost, and security. Nazari et al. [25] proposed a feature template protection scheme based on error correction codes and chaotic mapping. In this scheme, error correction codes are used to enhance the authentication ability of the template, and chaotic permutation is used to enhance the security and privacy of the template. The scheme has good resistance to brute force cracking and cross link attacks. Abou elazm et al. [26] proposed a reversible face template generation scheme based on 3D mosaic transformation and optical encryption. In this scheme, the extracted biometric features are first processed by bit-plane displacement, and then the template is generated by optical encryption using random phase mask. The scheme has good revocability and high recognition accuracy.

In application, templates may suffer from masquerade attack [27], spoofing attack [28], template reconstruction [29], and other threats. Once the template is cracked, it will seriously threaten the privacy and security of users. With the increasing demand for security, people are paying more and more attention to the security of templates. Determining how to establish a secure template and effectively protect the privacy of users has become an urgent problem to be solved.

## 3. Proposed Approach

### 3.1. Template Generation Scheme

In order to solve the problems mentioned in the previous section, this paper proposes a safe and effective face template protection method. The framework of the proposed scheme is shown in Figure 1. When the user registers the template, the face feature vector is extracted first, and then a chaotic sequence is used to scramble the feature vector to eliminate the correlation within the vector. Secondly, the chaotic sequence is used to generate an orthogonal matrix. The orthogonal matrix is used to change the state value of the feature vector. Finally, the chaotic sequence is used to construct a random matrix, and the cosine values of the included angle between the vector and the columns of the random matrix are calculated and converted to generate the template. The user registration process is described below.

**Step 1.** Extract the feature vectors of the face image.

Use feature extraction algorithm to extract feature vector of registered face image. Let *F* = [*f*_0_, *f*_1_, *f*_2_, ⋯, *f_k_*_−1_] denotes the extracted feature vector of the registered face image, where k represents the dimension of the extracted feature vector.

**Step 2.** Generate chaotic sequences.

Liu et al. proposed a chaotic system named coupled piecewise sine map (CPSM), it has large parameter space and good complexity, and the generated chaotic sequences of all dimensions have good randomness [30]. The randomness test for CPSM is shown in Appendix A. We use the three-dimensional form of CPSM to generate chaotic sequences, which is defined as
(1)$$\left\{\begin{array}{l} w=PSM(\lambda w+(1-\lambda )x)\\ x=PSM(\lambda x+(1-\lambda )y)\\ y=PSM(\lambda y+(1-\lambda )w) \end{array}\right.,$$
(2)$$\begin{array}{cc} x_{i+1} & =PSM(x_{i})\\ & =\left\{\begin{array}{ll} \sin(x_{i}\pi )/\sin(\mu \pi ), & \mathrm{if}\;\;x_{i}\leq \mu \;\;\;\mathrm{or}\;\;\;x_{i}>1-\mu ;\\ \sin((2\mu (x_{i}-\mu )/(1-2\mu ))\pi )/\sin(\mu \pi ), & \mathrm{if}\;\;\mu <x_{i}\leq 0.5;\\ \sin((1-\mu +(2x_{i}-1)\mu /(1-2\mu ))\pi )/\sin(\mu \pi ), & \mathrm{if}\;\;\;0.5<x_{i}\leq 1-\mu . \end{array}\right. \end{array}$$

Here, *λ* is the coupling parameter with a value range of [0.99, 1], and *μ* is used to control the piecewise interval with a value range of (0, 0.1]. The parameters of the scheme are used as the initial values of CPSM, and then generate three chaotic sequences $W=\{w_{i}\},X=\{x_{i}\},Y=\{y_{i}\},i=0,1,2,\cdots ,k^{2}-1$.

**Step 3.** Permute the feature vector.

This step is used to disturb the relationship between the dimensions of the vector. We use chaotic sequence *W* to permute the vector *F*. For the *i*-th dimension of *F*, we calculate the new position by
(3)$$j=\left\lfloor w_{i}\times 2^{16}\right\rfloor \;\;\;\mathrm{mod}\;\;k,$$
then swap the positions of *f_i_* and *f_j_*. The vector *F′* is obtained after scrambling *F*.

**Step 4.** Generate random matrix.

Because of the good randomness of chaotic sequence, the random matrix can be generated efficiently by using chaotic sequence. We use the chaotic sequence *X* to generate the random matrix. Let $R=\{r_{i,j}|i,j=0,1,2,\cdots ,k-1\}$ denote the generated random matrix, where $r_{i,j}=x_{i\times k+j}$.

**Step 5.** Generate orthogonal matrix.

Gram-Schmidt is an effective method for generating orthogonal matrices. We use this method to convert random matrix *R* into orthogonal matrix *G*.

**Step 6.** Transform the vector.

Transform the vector by $F^{\prime \prime }=F^{\prime }\cdot G$. Here, operator · represents dot product operation. This step performs dot product operation on the feature vector *F′* and each column of orthogonal matrix *G*.

**Step 7.** Generate random matrix.

Using the method in Step 4 to generate random matrix *R′* with chaotic sequence *Y*.

**Step 8.** Generate the template.

Let *R_i_* denote the *i*-th column in *R′*. For each column in *R′*, calculate the cosine values of the included angle between feature vector *F″* and column *R_i_* by
(4)$$S_{i}=\frac{F^{\prime \prime }\cdot R_{i}}{\left\|F^{\prime \prime }\right\|\left\|R_{i}\right\|}.$$

Here, operator $\left\|*\right\|$ is used to calculate the vector length. $S_{i}$ can accurately reflect the spatial position of *F″* and *R_i_*. Then, convert the cosine value of the floating-point type to an integer by
(5)$$t_{i}=\left\lfloor \frac{S_{i}}{\theta }\right\rfloor .$$

Here, $\left\lfloor *\right\rfloor$ stands for round down, *θ* is used to control the conversion, and 0 < *θ* < 1. Let $T=\{t_{i}|i=0,1,2,\cdots ,k-1\}$, *T* is the generated template.

### 3.2. Some Advantages of the Scheme

The proposed scheme has the following advantages:First, the chaotic system is used to drive the template generation process, which increases the revocable and diversity of the scheme. Taking advantage of the sensitivity of chaotic system to parameters, any slight modification of the key will produce completely different templates, making the templates generated by the scheme more diversified. When the template is leaked, different templates can be generated by changing the key, and the original template will be invalidated to ensure that the scheme has good revocability.Secondly, the orthogonal matrix is used to transform the vector value to ensure that the intermediate change process does not affect the verification performance. The orthogonal matrix is used for random projection of the feature vector, which can ensure that the transformed feature vectors have the distance preservation property, and the transformed feature vectors will not affect the verification performance. In addition, the orthogonal matrix is generated based on chaotic sequence, which not only has good generation efficiency, but also has good diversity.Finally, the cosine value of the included angle between the feature vector and the random vector is converted into integer data, which not only ensures the verification performance, but also makes the scheme irreversible. Calculating the cosine value of the angle between the feature vector and the column vector of the random matrix can accurately describe the relationship of the feature vector in the space formed by the random matrix. Although there is some information loss in converting the included angle cosine value into integer data, this defect can be remedied by combining the conversion of multiple included angle cosine values to ensure the verification performance. Moreover, the cosine value of the included angle does not have one-to-one correspondence with the integer data, which makes the scheme irreversible.

## 4. Experimental and Analysis

### 4.1. Experiment Setting

Dlib [31] is a modern toolkit that contains machine learning algorithms and tools for solving practical problems. It is widely used in industry and academia. Its face recognition subset can accurately calibrate and extract face features. This paper uses Dlib as the feature vector extraction tool. To verify the performance of the scheme, two face databases are used for testing: RaFD [32] and Aberdeen [33]. The RaFD databset contains 67 different individuals, and each individual contains 24 images with different expressions and gaze directions. The Aberdeen dataset has 687 color faces of 90 individuals, each individual has between 1 and 18 images. For some images, there are some variations in lighting and viewpoint. In the experiment, we set *θ* = 0.02.

### 4.2. Performance Verification

In order to verify the performance of the proposed scheme, 4000 pairs of images are randomly selected from each dataset. The images selected in each database include two images of 2000 pairs of the same person and two images of 2000 pairs of different people. For each pair of images, the corresponding templates are generated, and the cosine similarity is used to calculate the similarity of each pair of templates. According to the test results, we draw the receiver operating characteristic (ROC) curves, as shown in Figure 2. It can be seen that on the RaFD and Aberdeen datasets, the left part of each curve rises almost vertically to the top, and the upper region is very narrow, which means that the proposed scheme has good accuracy.

According to the above experiment, we calculated the equal error rate (EER) of the proposed scheme on two datasets. The EER of RaFD is 0.0055 and Aberdeen is 0.0045. The EER on different datasets are all very small, which means that the proposed scheme has good performance. To further demonstrate the performance of the scheme, the metrics listed in Table 1 are used to measure the scheme, and the results are shown in Table 2. We can see from Table 2 that the values of performance metrics in different datasets are very close to 1, indicating that the scheme has good verification performance. In addition, Table 3 lists the accuracy comparison between the proposed scheme and other state-of-the-art schemes. It can be seen that the proposed scheme has advantages in accuracy, compared with other schemes. These results show the effectiveness of the proposed scheme.

### 4.3. Privacy Analysis

#### 4.3.1. Irreversibility Analysis

When the template is irreversible, even if the attacker has both the key and the template, the original biometric information cannot be restored through the leaked information. After scrambling and transforming the extracted feature vector, the proposed scheme uses local sensitive hashes to generate template. The scheme calculates the similarity scores between the feature vector and the random matrix, and then delivers the scores according to the bucket. In this way, the high-dimensional feature vector is projected into an integer. This generation method is a one-way process and cannot be reversed. That is, the template can only be calculated from feature vector, and the feature vector cannot be restored from template. Therefore, even if the template generated by the scheme is leaked, it will not threaten the security of the user’s original facial feature information. In other words, the proposed scheme is irreversible.

#### 4.3.2. Revocability Analysis

In real application scenarios, template leakage is unavoidable. When using a revocable template, if the template being used in the authentication system is leaked, the template can be regenerated and make the leaked template invalidated. That is to say, different feature templates can be generated for the same person, and different templates cannot authenticate each other. The proposed scheme relies on the random number sequence generated by the chaotic system at all stages of vector permutation, vector transformation and template generation. Because CPSM is extremely sensitive to the initial parameters, any slight change in the initial parameters will produce completely different random number sequences. Using different random number sequences will lead to completely different states of the generation process, and then completely different feature template will be obtained. In other words, different templates can be generated by changing the parameters of the chaotic system.

In order to test the revocability of the proposed scheme, we tested it on two datasets. In each dataset, we randomly selected one image of each individual as the registered image and used 20 different sets of keys to generate 20 different sets of templates; then, we performed the following two queries:Genuine: For the same person, use different image to query in the template database.Mated imposter: For the same person, calculate the difference between different template databases.

For the two queries on each dataset, we calculated the similarity score by using cosine similarity. Figure 3 is the probability distribution diagram drawn according to the query results. It can be seen from the figure that on the RaFD and Aberdeen datasets, the similarity scores of genuine queries are concentrated in the range above 0.8, while the similarity scores of mated imposter queries are concentrated in the range below 0.4. The similarity scores of the two queries are significantly different. The results show that, when the same user registers in the systems with different keys, the generated templates are significantly different. When a feature template is leaked, different keys can be used to generate new templates. That is, the proposed scheme has good revocability.

#### 4.3.3. Unlinkability Analysis

When the feature template is non-linkable, the feature template generated by the same user using different keys cannot be matched successfully in another authentication system. We verified the unlinkability of the scheme on the template databases generated in the previous subsection. For each dataset, we calculated the similarity scores of mated imposter query and non-mated imposter query between templates generated with different keys. The calculation objects of non-mated imposter query are different users in different template databases. Figure 4 is the probability distribution diagram of similarity scores, drawn according to the calculation results. It can be seen from the figure that the distribution curves of the similarity scores of the mated imposter query and non-mated imposter query of each dataset have a high degree of coincidence. According to the results of genuine queries and mated imposter queries in previous subsection, when a record in one template database is used to match in another template database, it cannot be verified, and according to the similarity score of the match, it cannot be determined whether the user corresponding to the record is registered in another template database. The experimental result demonstrates that the proposed scheme is non-linkable.

### 4.4. Security Analysis

#### 4.4.1. Key Space Analysis

The scheme used the chaotic sequences that were produced by the CPSM to drive the template generation process. For the CPSM is very sensitive to the initial parameters, different parameters will produce different chaotic sequences that generate different templates. Therefore, the parameter space of the CPSM can be regarded as the key space of the proposed scheme. The proposed scheme uses the three-dimensional form of CPSM, and its parameter includes *w*, *x*, *y*, *μ* and *λ*. The value range of *w*, *x*, *y* is [0, 1], *μ* is [0, 0.1] and *λ* is [0.99, 1]. According to IEEE standard [37], in 64-bit computers, the precision of floating-point number is 10^−15^. We can get that the key space of the proposed scheme is $10^{15}\times 10^{15}\times 10^{15}\times (0.1\times 10^{15})\times (0.01\times 10^{15})\approx 2^{239}$. When using the proposed scheme, the same user can register 2^239^ different templates in different systems. The diversity of templates can significantly improve the ability to resist brute force attacks.

#### 4.4.2. Hill Climbing Attack Analysis

In the hill climbing attack scenario, an attacker wants to impersonate a user to pass the verification of the system. He can submit the attack data and obtain the matching score from the verification system. According to the matching score, the attacker constantly adjusts the submitted attack data until it passes the verification [38]. The hill climbing attack does not require the attacker to have any prior knowledge—that is, the attacker does not know the template generation process and the matching score calculation method. In the proposed scheme, it is assumed that the attacker submits a feature vector for query and obtains matching scores. Then, the attacker tries to adjust the value of the feature vector constantly to pass the verification. According to the generation process of the template, when any dimension of the input face feature vector is changed, the value of each dimension of the generated template may change. Therefore, it is difficult for the attacker to determine the influence of each dimension of the input face feature vector on the final query result. The most effective way for an attacker to successfully deceive the verification system is to constantly try the combination of all feature vectors. Assuming that the value of each dimension of the feature vector has 4 possibilities, and the dimension of the feature vector is 128, there are 2^256^ possibilities for all combinations of the feature vector. As a matter of fact, each dimension of feature vector has more than four values, so it is difficult to implement a hill climbing attack to pass verification.

#### 4.4.3. Lost Template and Lost Key Attack

Once a malicious attacker obtains the template database and the key to a verification system, he hopes to restore the original feature vector of a user through the obtained data. Assuming that the attacker knows the template generation method, the attacker can master the detailed template generation process after obtaining the key. However, it has been shown in irreversible analysis that the proposed scheme is irreversible. Therefore, even if the attacker has mastered the template database and the corresponding key, the original face feature vector cannot be restored by the reverse process of generating the feature template.

#### 4.4.4. Attacks via Record Multiplicity

In the attacks via record multiplicity (ARM), for a user, an attacker obtained multiple different templates from different verification systems. The attacker attempts to restore a possible pre-image of the user using the correlation between different templates. The experimental results of unlinkability analysis show that there is no significant difference between the matching results of the same user and the matching results of different users on different template databases. That is, the attacker cannot obtain useful information through cross-matching of different templates. In addition, when different keys are used, the permutation order, orthogonal matrix, and random matrix of the generation process are completely different, which makes the *i*-th dimension of the face feature vector and the *i*-th dimension of the template have no direct correlation, and the same dimension of different templates has no direct correlation. This makes it impossible to derive useful information from each other. As a result, the attacker cannot perform an ARM attack.

### 4.5. Running Speed Analysis

The most common application form of face recognition system is identification mode. When a user queries in the face recognition system, the query template is generated first, and then the template is matched with the records in the template database. The identification mode is a one-to-many comparison. The running speed of identification mode is related to the number of records in the template database. The more records, the slower the running speed.

We implemented the algorithm using Python 3.7 on a laptop with Intel Core i7-4710MQ @2.5GHz CPU and 8GB RAM, and selected one image of each person in RaFD and Aberdeen as the registered image to generate the template database. Then, we randomly selected non-registered images as query images to test the running speed. We conducted 2000 queries and calculated the average running time. The running time of the proposed scheme consisted of three parts: the first is the time to extract the face feature vector using Dlib, the second is the time to generate the query template, and the third is the time to compare in the template database. The first two items are relatively fixed, and the latter is related to the number of records in the template database. The test results are shown in Table 4. It can be seen that, when the number of records in the template database is 157, the average query speed is 126.48 milliseconds, showing a fast running speed.

## 5. Conclusions

Aiming at template protection in face recognition system, this paper proposes a secure and effective template generation scheme based on a chaotic system. The template generation process includes vector permutation, vector transformation, angle cosine calculation, and conversion. The steps of the generation process depend on the chaotic sequences generated by the chaotic system. Because the chaotic system is highly sensitive to the initial parameters, the scheme can easily generate different templates with different keys and make different the templates have good differentiation. The scheme converts the cosine values of the included angle between the feature vector and different random vectors into integers to generate the template, which makes the template irreversible and can significantly improve the security. The experimental results on different datasets prove that the scheme has good verification performance and efficiency. Privacy analysis and security analysis show that the scheme can resist various common attacks and effectively ensure the security of the template.

The proposed scheme only uses system parameters to control template generation, which has some limitations. In extreme cases, for example, a manager manages several different face recognition systems at the same time, but he uses the same parameters in these systems. If a user has registered in these systems, the templates in different systems can be cross-referenced. This will significantly reduce the security of the scheme. In the future, we consider that users can set their own parameters at the template registration stage. When a user registers in different systems, the above problems can be avoided by setting different user parameters. How to store user parameters safely is the key problem we need to solve.

## Figures and Tables

**Figure 1 entropy-25-00378-f001:**
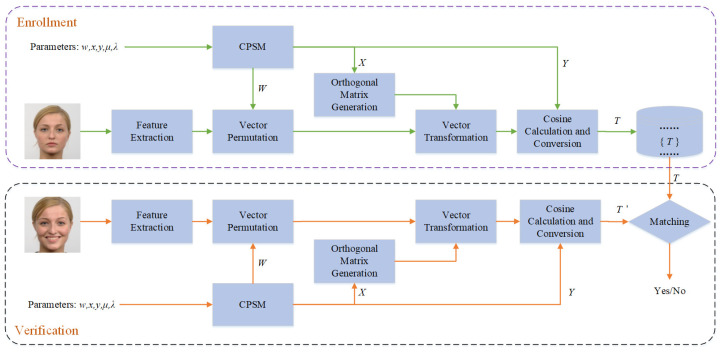
The framework of proposed scheme.

**Figure 2 entropy-25-00378-f002:**
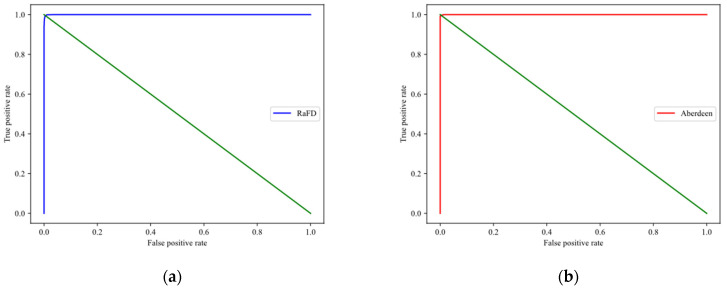
ROC curves on different datasets: (**a**) RaFD; (**b**) Aberdeen.

**Figure 3 entropy-25-00378-f003:**
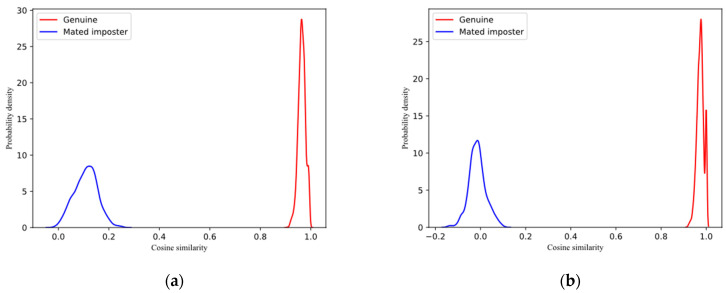
Similarity scores of genuine and mated imposter: (**a**) RaFD; (**b**) Aberdeen.

**Figure 4 entropy-25-00378-f004:**
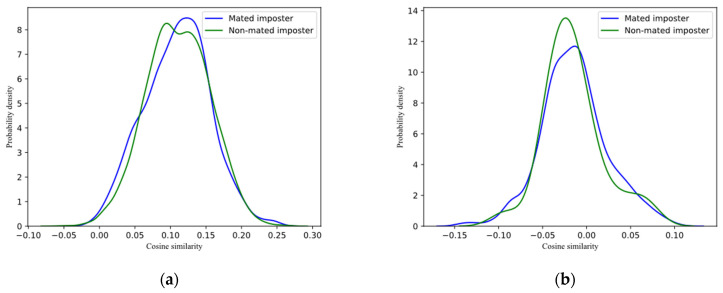
Similarity scores of mated imposter and non-mated imposter: (**a**) RaFD; (**b**) Aberdeen.

**Table 1 entropy-25-00378-t001:** Equations of the performance metrics.

Performance Metric	Equation
*Accuracy*	$\frac{TP+TN}{TP+FP+FN+{TN\;}^{1}}\times 100\%$
*Specificity*	$\frac{TN}{FP+TN}\times 100\%$
*Precision*	$\frac{TP}{TP+FP}\times 100\%$
*Recall*	$\frac{TP}{TP+FN}\times 100\%$
*F_score_*	$\frac{2\times Recall\times Precision}{Recall+Precision}\times 100\%$

^1^ True positive (TP) is successfully verified with real data, false positive (FP) is successfully verified with false data, false negative (FN) is the use of real data validation failed, and true negative (TN) is the use of false data validation failed.

**Table 2 entropy-25-00378-t002:** Performance metrics of the scheme.

Dataset	*Accuracy* (%)	*Specificity* (%)	*Precision* (%)	*Recall* (%)	*F_score_* (%)
RaFD	99.40	99.45	99.44	99.35	99.40
Aberdeen	99.63	99.55	99.55	99.70	99.63

**Table 3 entropy-25-00378-t003:** Comparison between the proposed scheme and others.

Method	Dataset	*Accuracy* (%)
Gradient RP-Q2DPCA [34]	Aberdeen	97.70
SPGPFL [35]	Aberdeen	97.30
Weighted Intensity PCNN [36]	Aberdeen	96.00
Proposed	Aberdeen	99.63

**Table 4 entropy-25-00378-t004:** Running speed test of the scheme (ms).

Feature Vector Extraction	Query Template Generation	Comparison
73.58	32.61	20.29

## Data Availability

Not applicable.

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
