# Peer review of "An Irreversible and Revocable Template Generation Scheme Based on Chaotic System"

_entropy, 2023, doi:10.3390/e25020378_

Round 1

Reviewer 1 Report

In this paper, a novel irreversible and revocable template generation scheme based on chaotic system is proposed, Privacy analysis and simulation results are given to verify its effectiveness. It is interesting. But I have some comments:

1. The advantages of the proposed work should be highlighted in the abstract and conclusion.

2. The first condition of formula (2) seems to have a writing error, please check and confirm it.

3. In line 175, please explain the operational meaning of the "*" symbol.

4. The symbols appearing in the numerator and denominator of formula (4) shall be explained.

5. Except for the security, running time is also important. The running time of the proposed scheme is needed.

Author Response

Point 1: The advantages of the proposed work should be highlighted in the abstract and conclusion.

Response 1: Thanks. In the revised abstract and conclusion, we emphasized the role of chaotic system on template and the characteristics of the scheme. At the same time, in the introduction section we have listed the novelty and main contributions of this paper.

Point 2: The first condition of formula (2) seems to have a writing error, please check and confirm it.

Response 2: Thanks. The first condition of formula (2) means xi≤µ or xi>1-µ . To make the formula clearer, we have corrected writing.

Point 3: In line 175, please explain the operational meaning of the "*" symbol.

Response 3: Thanks. The operator "•" represents dot product operation. We have explained it in the revised manuscript.

Point 4: The symbols appearing in the numerator and denominator of formula (4) shall be explained.

Response 4: Thanks. The operator "||*||" is used to calculate the vector length. We have explained it in the revised manuscript. For the operator "•"  has been explained in Step 6, so it is not explained here again.

Point 5: Except for the security, running time is also important. The running time of the proposed scheme is needed.

Response 5: Thanks. We have added the subsection of running speed analysis. We implemented the algorithm using Python 3.7 and analyzed the running speed of the scheme using identification mode. In the proposed scheme, the identification process includes three steps: Feature vector extraction, Query template generation and comparison. We analyze the running time from these aspects. The results show that the scheme has a fast running speed. Please see Section 4.5 for details.

Reviewer 2 Report

The paper proposes a secure image template generation scheme based on CPSM chaotic system. The paper is interesting and generally well written. However, some issues need to be solved:

1.       The abbreviation CPSM needs to be written in full the first time it is used;

2.       On what basis the CPSM chaotic system was selected? Some details are needed. Can we use another chaotic system instead of CPSM?

3.       Figure 1: what the most-right block from Enrollment (the one with … {T} …) represents?

4.       How the five CPSM parameters leakage affect the security of the entire scheme? How are they stored/transmitted to prevent their leakage?

5.       Some sentences need to be rewritten: a) lines 36-39: the description of “Irreversibility” is very long and, because of this, it is somehow confusing; b) lines 47-48: the sentence “The template …” is confusing; c) line 79: what “strong invisibility” means? d) lines 81-82: the sequence “into a predefined hash value” is confusing; e) line 132: The sequence “In the actual environment” is confusing in the context. What “environment” means in the context? f) line 322: it is written “template” instead of “templates”;

6.       Lines 152-184: Some sentences (see lines 162-163 and 165-166 for example) look like lines from a pseudocode (they are not integrated in the context) but this part is not actually a pseudocode. Please reshape.

Author Response

Point 1: The abbreviation CPSM needs to be written in full the first time it is used

Response 1: Thanks. CPSM is the abbreviation of Coupled Piecewise Sine Map. We have corrected it in the paper.

Point 2: On what basis the CPSM chaotic system was selected? Some details are needed. Can we use another chaotic system instead of CPSM?

Response 2: Thanks. CPSM is one of our work, published in "Nonlinear dynamics". The chaotic sequence generated by CPSM has good randomness and uniform distribution. To further demonstrate the performance of the chaotic system, we used NIST to test the generated chaotic sequence in Appendix A. The test results proved the excellent performance of the system. The use of chaotic systems with good performance is conducive to improving the security of the scheme, such as a larger parameter space and a stronger ability to resist attacks. Of course, other chaotic systems with excellent performance can instead of CPSM.

Point 3: Figure 1: what the most-right block from Enrollment (the one with … {T} …) represents?

Response 3: Thanks. In the Figure 1, the “T” denotes the generated template. We used “{T}” to indicate that “T” is a record in the template database.

Point 4: How the five CPSM parameters leakage affect the security of the entire scheme? How are they stored/transmitted to prevent their leakage?

Response 4: Thanks. If the parameters are leaked, the attacker can generate the same chaotic sequence, thus mastering the detailed process of template generation. However, the template generation process is irreversible. Even if the attacker knows the vector scrambling order, orthogonal matrix and random matrix, it cannot restore the feature information of registered users. The application scenario of the scheme we designed is assumed to be a single-node host mode, that is, the face image acquisition and authentication processes are implemented on a host. In this mode, the parameters can be stored in the host. At present, we are studying the face authentication process in the non-secure network scenario. In this mode, we need to solve the problems of parameter transmission, feature information transmission, etc.

Point 5: Some sentences need to be rewritten: a) lines 36-39: the description of “Irreversibility” is very long and, because of this, it is somehow confusing; b) lines 47-48: the sentence “The template …” is confusing; c) line 79: what “strong invisibility” means? d) lines 81-82: the sequence “into a predefined hash value” is confusing; e) line 132: The sequence “In the actual environment” is confusing in the context. What “environment” means in the context? f) line 322: it is written “template” instead of “templates”;

Response 5: Thanks. We have corrected the description of some sentences to make them more concise, and also corrected some grammatical errors.

Point 6: Lines 152-184: Some sentences (see lines 162-163 and 165-166 for example) look like lines from a pseudocode (they are not integrated in the context) but this part is not actually a pseudocode. Please reshape.

Response 6: Thanks. We have changed some descriptions of the template generation process to make it more accurate. Please see subsection 3.1 for details.

Reviewer 3 Report

This paper proposes a secure template generation scheme for face recognition technology based on chaotic systems. The scheme involves several steps, including permuting the extracted face feature vector to eliminate correlations, transforming the vector using an orthogonal matrix, calculating the cosine value of the included angle between the feature vector and different random vectors, and converting the results into integers to generate the template. In terms of the strengths of the paper, the authors have clearly outlined the steps involved in their proposed template generation scheme and provided experimental results to support its effectiveness. They have also discussed the security and privacy properties of their scheme and demonstrated how it can resist common attacks.

However, there are some potential limitations of the paper. One aspect that could be improved is a more in-depth explanation and justification of the choice of chaotic systems as the basis for the template generation process. Additionally, a more comprehensive evaluation and comparison with existing face recognition template generation methods and various chaotic systems such as presented in “A novel approach for synchronizing of fractional order uncertain chaotic systems in the presence of unknown time-variant delay and disturbance”, “A symmetric key multiple color image cipher based on cellular automata, chaos theory and image mixing”, and “An image encryption scheme based on block scrambling, modified zigzag transformation and key generation using enhanced logistic-tent map” would strengthen the validity of the authors' claims. Futher issues: The novelty and contribution of this paper is not stated. The proposed chaotic system is not evaluated using the NIST tests for randomness. The limitations of the proposed approach are not discussed.

In conclusion, the paper presents a novel approach to secure template generation in face recognition systems using chaotic systems. While it has some strengths, more in-depth statistical analysis and evaluation is necessary to fully assess its effectiveness and practical implications.

Author Response

Point 1: One aspect that could be improved is a more in-depth explanation and justification of the choice of chaotic systems as the basis for the template generation process.

Response 1: Thanks. In the Introduction, our revised content stated the advantages of generating templates based on chaotic systems: it can ensure the revocability and diversity, and it is beneficial to improve security and make the generation process more convenient. For any chaotic system with excellent characteristics, it can be selected to drive the generation process. CPSM is one of our work, published in "Nonlinear dynamics". The chaotic sequence generated by CPSM has good randomness and uniform distribution. So we selected CPSM as the chaotic system. In subsection 3.1, we added some descriptions about why choose CPSM.

Point 2: Additionally, a more comprehensive evaluation and comparison with existing face recognition template generation methods and various chaotic systems such as presented in “A novel approach for synchronizing of fractional order uncertain chaotic systems in the presence of unknown time-variant delay and disturbance”, “A symmetric key multiple color image cipher based on cellular automata, chaos theory and image mixing”, and “An image encryption scheme based on block scrambling, modified zigzag transformation and key generation using enhanced logistic-tent map” would strengthen the validity of the authors' claims.

Response 2: Thanks. In the introduction, we analyzed from the perspective of these literatures. These documents provide some new references.

Point 3: Futher issues: The novelty and contribution of this paper is not stated.

Response 3: Thanks. We have added a description of the novelty and contribution of the scheme. Please see introduction for details.

Point 4: The proposed chaotic system is not evaluated using the NIST tests for randomness.

Response 4: Thanks. In order to prove the excellent performance of CPSM, NIST was used for testing, and the test results are shown in Appendix A. CPSM has passed various tests of NIST, indicating that the generated chaotic sequence has good randomness.

Point 5: The limitations of the proposed approach are not discussed.

Response 5: Thanks. Only using system parameters to control feature template generation has certain limitations. If two different face authentication systems use the same system parameters, the security of the scheme will be reduced. This is a common problem in most current schemes. We consider adding user parameters when users register to further improve the security of the scheme. This is our next research work. In the conclusion, we add the prospect of future research.

Point 6: In conclusion, the paper presents a novel approach to secure template generation in face recognition systems using chaotic systems. While it has some strengths, more in-depth statistical analysis and evaluation is necessary to fully assess its effectiveness and practical implications.

Response 6: Thanks. In order to further prove the effectiveness of the scheme, we added a subsection of running speed analysis. When the proposed scheme is used as identification mode, the runtime includes three parts: first, feature extraction, second, query template generation, and third, matching. We have tested the three parts of the time, and the results show that the proposed scheme has a fast running speed, which proves that the proposed scheme has a good running efficiency. Please see Section 4.5 for details.

Round 2

Reviewer 2 Report

The authors have successfully solved all y comments and concerns.

Author Response

Thank you very much for your efforts.

Reviewer 3 Report

The authors have revised well. I have only a few technical comments:

- Abstract: mention the datasets used in this study.

- Table 1: the equations are not calculated in percents although the values in Table 2 are given in percents.

- Discuss the limitations of the proposed scheme.

Author Response

Point 1: Abstract: mention the datasets used in this study.

Response 1: Thanks. In the introduction of the revised version, we mentioned that the datasets used in the article are RaFD and Aberdeen.

Point 2: Table 1: the equations are not calculated in percents although the values in Table 2 are given in percents.

Response 2: Thanks. We corrected the writing of the formulas in Table 1. We will pay attention to avoid similar problems in the future.

Point 3: Discuss the limitations of the proposed scheme.

Response 3: Thanks. We revised the conclusion and pointed out the limitations of the scheme: if different systems use the same parameters, the security of the scheme will be reduced. We also explain some new research ideas in the future: solve the problem by adding user parameters. This method needs to solve the problem of safe storage of user parameters and consider the convenience of use. This is our current research. Please see the conclusion for details.